# LPI Radar Waveform Recognition Based on Features from Multiple Images

**DOI:** 10.3390/s20020526

**Published:** 2020-01-17

**Authors:** Zhiyuan Ma, Zhi Huang, Anni Lin, Guangming Huang

**Affiliations:** 1College of Physical Science and Technology, Central China Normal University, No.152 Luoyu Road, Wuhan 430079, China; mazhiyuan@mails.ccnu.edu.cn; 2Department of Electronic Technology, Naval University of Engineering, Wuhan 430033, China; HzpaperPHD@163.com (Z.H.); Linanni_ME@163.com (A.L.)

**Keywords:** low probability of intercept, multiple feature image, short-time autocorrelation

## Abstract

Detecting and classifying the modulation type of the intercepted noisy LPI (low probability of intercept) radar signals in real-time is a necessary survival technique in the electronic intelligence systems. Most radar signals have been designed to have LPI properties; therefore, the LPI radar waveform recognition technique (LWRT) has recently gained increasing attention. In this paper, we propose a multiple feature images joint decision (MFIJD) model with two different feature extraction structures that fully extract the pixel feature to obtain the pre-classification results of each feature image for the non-stationary characteristics of most LPI radar signals. The core technology of this model is combining the short-time autocorrelation feature image, double short-time autocorrelation feature image and the original signal time-frequency image (TFI) simultaneously input into the hybrid model classifier, which is suitable for non-stationary signals, and it has higher universality. We demonstrate the performance of MFIJD by simulating 11 types of the signals defined in this paper and generating training sets and test sets. The comparison with the literature shows that the proposed methods not only has a high universality for LPI radar signals, but also better adapts to LPI radar waveform recognition at low SNR (signal to noise ratio) environment. The overall recognition rate of the method reaches 87.7% when the SNR is −6 dB.

## 1. Introduction

The low probability of intercept (LPI) radar waveform recognition is one of the key technologies in the radar countermeasure system, and it is also an important content in radar signal processing. In practice, the automatic radar waveform recognition technique is a core survival technique for an intercept receiver performing threat recognition and radar emitter identification [1]. On the contrary, the radar signals should be designed to have LPI features, so that the radar signal is not easily detected and recognized by the intercepting receivers. Therefore, intercept receivers should be necessarily equipped with an automatic LPI radar waveform recognition function that provides highly reliable detection, classification, and identification capabilities in order to recognize the presence of the LPI radar signal in advance [2].

It is difficult to obtain a satisfactory recognition rate based on conventional parameter methods [3,4], Because LPI radar signals generally adopt complex intra-pulse modulation methods and pulse compression transmission systems, such as linear frequency modulation (LFM), Costas codes, polytime codes (T1, T2, T3, and T4), and polyphase codes (comprising Frank, P1, P2, P3 and P4), frequency diversity and frequency agility in complex and variable electronic countermeasure environments. At present, there are two main problems in the classification of LPI radars to be solved. For one thing, some existing feature extraction methods are highly targeted, mainly for some specific radar emitter signals. For another thing, these methods rarely involve problems with noise effects and low SNR. In fact, radar signals, especially LPI radar signals, are inevitably subject to large amounts of noise during propagation and reception [5], and LPI radars usually have lower power and it is difficult to directly classify [6]. In current research, there have been some LPI waveform recognition technologies (LWRT), which use feature extraction and classification techniques. Time-frequency analysis (TFA) is widely used in the feature extraction since LPI radar signals are usually non-stationary signals, such as Smoothed Pseudo-Wigner Distribution (SPWD) [7], Wigner Ville Distribution (WVD) [8], Short-Time Fourier Transform (STFT) [9,10,11], and Choi-Williams Distribution (CWD) [6,12,13,14,15,16]. Combined with deep learning in the field of computer vision [17] and models of neural network structures, researchers have obtained better recognition results from the time-frequency feature of signals [18]. The radar signal is first time-frequency transformed into a two-dimensional time-frequency image (TFI), which then is preprocessed and sent to a neural network for training. In the area of classifier design, classification methods include multi-Layer Perceptron (MLP) [11], conditional decision for different features [11,15], Convolutional Neural Networks (CNN) [14], Elman Neural Networks (ENN) [6], and support vector machines (SVM) [6,16]. In addition, there have been hybrid classifiers designed that combine a few different multiple classifiers. For instance, in [15,19,20], all of them combine two deep learning models. In [15], the classifier consists of CNN and ENN. In [20], the classifier consists of CNN and SAE. 

However, the feature of TFI will be overwhelmed by a large amount of noise due to the interference of noise. Researchers have dedicated their attention to image pre-processing, generally in two ways, in order to solve this problem. First, before time-frequency conversion, algorithm filtering is used to reduce noise [21,22,23,24]; second, some image processing technology is used to process TFI to achieve accurate classification [6,13,15,16]. For the first method, most of the current noise reduction algorithms are difficult to apply in a low SNR environment. The digital average denoising [14] did not further analyze how to achieve the trigger of digital average at low SNR. For the TFI that is generated by adaptive filtering combined with various LMS algorithms [23,24], the noise component will completely overwhelm the signal component. In [25], the feature image construction algorithm can perform autocorrelation processing on only six types of conventional radar signals (CP, LFM, NCPM, BPSK, BFSK, QFSK), but it is not applicable to non-stationary signals, such as Costas, LFM, P1-P4, T1-T4, and Frank. For the second method, literature [6,12,13], and [16] reduce the TFI noise through image processing, but the processing methods of image morphology and threshold filtering will easily lose the signal components. In addition, [6,13] and [16] only studied the classification of 8 types of LPI radar signals, the polytime codes (T1, T2, T3, and T4) are not mentioned in [6,16], the polyphase codes (P1, P2, P3 and P4) are not mentioned in [13]. Furthermore, the recognition rate under the SNR of −4 dB has not been studied in [6,13,16]. Obviously, in order to obtain better classification results, it is necessary to design a receiver capable of classifying more types of radar signals at a lower SNR.

In this paper, we propose a new signal preprocessing technique by using short-time autocorrelation and combine TFA to solve the problem of the low recognition rate of multiple types of LPI radar signal for classification. In addition, we propose a multiple feature images joint decision (MFIJD) model, in which the multiple feature images different from the traditional TFI are generated, while combining with network and inference machine module. Three groups of parallel depth models are used simultaneously to better extract the deep features of the TFI. Afterwards, three TFIs, including original TFI, short-time autocorrelation TFI and multiple short-time autocorrelation TFI are input to the MFIJD model, thereby obtaining a pre-classification result for each feature image. At the same time, two different structures for feature extraction including CNN and BiLSTM are designed and the CNN consists of 11 layers, including convolutional layer, pooling layer, and fully connected layer. The simulation results show that the proposed MFIJD model can classify twelve types of LPI radar signals when the SNR is −6 dB, the overall recognition rate of the method reaches 87.7%.

This paper is organized, as follows. Section 2 introduces the basic framework of radar signal waveform recognition and describes the LPI radar waveforms that were considered in this paper. Section 3 introduces the proposed pre-processing technique and proposes short-time autocorrelation feature image construction technique. A classifier is designed in Section 4 according to the short-time autocorrelation image, TFI, and the characteristics of multiple non-stationary signals, which can analyze the features of the signals to classify the radar waveform considered in this paper. In Section 5, the performance of the proposed model is compared to the recent LPI radar waveform recognition techniques. Finally, Section 6 draws the conclusion of this paper.

## 2. System Structure and Waveforms 

In this section, we present the system structure of the proposed LWRT and the definition of the twelve types of LPI radar waveforms that were considered in this paper.

### 2.1. Proposed LWRT

Nowadays, the methods that are based on the deep learning have a better performance in radar waveform recognition [14,21]. The overall block diagram of LWRT proposed in this paper is shown in Figure 1. The Radio Frequency (RF) signal is received into the signal receiver through the antenna, which is mixed and sampled in it. The intermediate frequency (IF) signal y(k) output from the signal receiver and they then are preprocessed to reduce the effect of spatial noise on the recognition. In this paper, the signal is transmitted in two channels. The signal of one channel, which is the original signal, is directly time-frequency transformed to generate TFI. The other channel signal is preprocessed first, and the signal is time-frequency transformed to generate two images after short-time autocorrelation and double short-time autocorrelation processing, respectively, which are called feature images. Subsequently, these three images are input to the pre-designed and pre-trained classifier to classify the signal modulation mode at the same time. The internal structure of the classifier is discussed in detail in Section 4.1. To stress that, this paper focused on the modulation method similar to [2,6,13,26] and assumes that we sample a complete data in one pulse of the signal.

### 2.2. LPI Radar Waveform 

The signal from the receiving module can be generally expressed, as below:(1)$$y(k)=s(k)+w(k)=A(k)e^{j\theta (k)}+w(k)$$
where, s(k) is the ideal discrete signal coming after IF sampling, w(k) is additive white Gaussian Noise (AGWN), k is the index value increasing with sampling interval sequentially, A(k) is the ideal sampling signal of instant envelope, θ(k) is the instantaneous phase of the ideal sampling signal, and instantaneous phase θ(k) can be computationally expressed further by instantaneous frequency f(k) and instantaneous phase offset ϕ(k) as below:(2)$$\theta (k)=2\pi f(k)(kT_{s})+\phi (k)$$
where, $T_{s}$ represents the signal sampling interval time. In practice, radars typically use frequency modulation (FM), phase modulation (PM), and amplitude modulation (AM) to form different transmit waveforms. A and $f_{c}$ represent constant amplitude and frequency respectively, ϕ(k) represent the instantaneous phase offset of the signal. It is difficult to realize waveform classification since the LPI signal modulation form is similar. Therefore, this paper focuses on the classification of LPI signals. In this subsection, we define the twelve LPI radar waveforms that were considered in this paper, as shown in Table 1. There are LFM, Costas code, five polyphaser codes (such as Frank, P1, P2, P3, and P4 codes), and four polytime codes (such as T1, T2, T3, and T4 codes) introduced in the literature [14].

In Table 1, M is the number of encoded phases, i, j represent an iterative integer value from 1 to M, and ρ denotes the number of coded phases. However, unlike M, ρ must take the value that can be squared, $N_{ps}$ represents the number of phase states, $N_{si}$ represents the number of step frequency segments, $\left\lfloor \alpha \right\rfloor$ represents the largest integer less than or equal to α, and $\mathrm{mod}\left(•\right)$ represents the remainder function.

## 3. Signal Pre-Processing

In this section, the time-domain waveform and time-frequency characteristics of LPI radar signals are analyzed. A method of processing the signals by using short-time autocorrelation combined with TFA is proposed to generate short-time autocorrelation images that are more adaptable for non-stationary signals.

### 3.1. TFA Technique for CWD-TFI

The transformation used in this paper is the Choi-illiams distribution (CWD) [14,21]. When applied to a signal *x*(*t*), the time-frequency transformation with bilinear form is expressed as
(3)$$C_{x}(t,\Omega |g)=\frac{1}{2\pi }∭x(u+\tau /2)x^{*}(u-\tau /2)g(\theta ,\tau )\;\;e^{-j(\theta t+\Omega \tau -u\theta )}dud\tau d\theta$$
where g(θ,τ) is the kernel function of TFI. By changing the parameters in g(θ,τ), we can acquire different time-frequency distributions.

In many TFA methods, by adjusting the parameter values, the CWD that is based on the nonlinear time-frequency representation can suppress the time-frequency cross-term [16], the kernel function of CWD is
(4)$$g(\theta ,\tau )=e^{-j\theta^{2}\tau^{2}/\sigma }.$$

Finally, the expression for CWD is
(5)$$C_{x}(t,\Omega )=\iint \sqrt{\frac{\pi \sigma }{\tau^{2}}}x(u+\tau /2)x^{*}(u-\tau /2)\;e^{-\pi^{2}\sigma {\left(u-t\right)}^{2}/4\tau^{2}-j\Omega \tau }dud\tau .$$
where $C_{x}(t,\Omega )$ is the result of TFT [22], t and Ω are the time and frequency axis, respectively, and the scale factor σ is used to control the distribution of CWD cross-terms. The frequency resolution decreases when σ inhibits the cross-terms in CWD. The value σ=1 is used to balance the relationship between CWD cross-term suppression and the frequency resolution. The Choi-Williams transform is mathematically expressed as $CWD\left(•\right)$. Then, for an input signal y(k), the result of TFT can be described as
(6)[t,f,trf]=CWD(y(k)).
where, t∈ℝn is a one-dimensional vector, f∈ℝn×n and trf∈ℝn×n are two-dimensional vectors. Each in f represents the frequency and each element in trf represents the signal strength. Each element in t represents the time, each element in f represents the normalized frequency value of the point, and each element in trf represents the strength of the point. The time-frequency diagram of the signal can be obtained after time-frequency analysis.

For convenience of expression, $C_{ij}$ represents the data in the two-dimensional matrix $C_{R}(i,l)$ of row i and column j, $C_{R}$ represents $C_{R}(i,l)$, and the equation can be written as $R_{y}(t)$
(7)$$Picture=255\frac{C_{ij}-\min(C_{R})}{\max(C_{R})-\min(C_{R})}$$
where $\max\left(•\right)$ and $\min\left(•\right)$, respectively, represent the maximum and minimum points of the two-dimensional matrix, and Picture is the feature image of various signals constructed in this paper that can represent the two-dimensional image formed by mapping the two-dimensional matrix to the pixel interval. According to TFA, the TFI of the signals in Table 1 are simulated in a noise-free environment. For convenience of explanation, this section uses single frequency signal, LFM, and P1 as an example analysis, as shown in Figure 2.

However, the TFI of the signal is susceptible to window function selection. Two window functions are selected in this paper, namely Kaiser window and Hamming window, in order to effectively extract the time-frequency feature of the signal. For the Kaiser window, its window function can be expressed as
(8)$$w(n)=\left\{\begin{array}{cc} \frac{I_{0}\beta \sqrt{1-{\left(\frac{2n}{N-1}-1\right)}^{2}}}{I_{0}\beta }, & 0\leq n\leq N-1\\ 0, & otherwise \end{array}\right.$$
where β is used to adjust window function performance, N is the window length, and $I_{0}$ is the first type of zero-order Bessel function. The TFI of signal generated by using the Kaiser window, as shown in Figure 3, where β=3π.

For the Hamming window, its window function can be expressed as
(9)$$w(n)=\left\{\begin{array}{cc} 0.54-0.46\cos(\frac{2\pi n}{N-1}), & 0\leq n\leq N-1\\ 0, & otherwise \end{array}\right.$$

For the two window functions, this paper simulates the TFIs that are generated by the SNR of 0 dB and 9 dB, respectively. Figure 3 shows the feature images generated by the Kaiser window and the Hamming window.

It can be found from Figure 3 that no matter which window function is selected, when the SNR is reduced, it has a great effect on the TFI. However, different window functions has different effects on the image. For the Kaiser window, the TFI characterizes the signal with a larger range of pixel features, which will be more conducive to signal feature extraction. However, its anti-noise ability is weak at low SNR and the TFI is greatly affected by noise, such as the signal P1, which is difficult to distinguish. When comparing the Kaiser window, it can be found that the pixel features of the TFI are less when the Hamming window is used. However, the pixel features are not susceptible to noise at low SNR, which will be more conducive to signal classification. Regarding the selection of window functions, this paper will explain in detail in Section 3.2.

It can be seen from Figure 3 that the TFI is susceptible to noise. Especially at low SNR, the TFI is almost indistinguishable, as shown in Figure 4a. The literature [6,15,16] preprocesses the TFI by threshold denoising in order to reduce the effect of noise on TFIs. However, this method has no effect at lower SNR. If the time-frequency image is directly used, the effect of noise cannot be avoided, which results in poor classification of the signal at low SNR. Therefore, in this paper, based on time-frequency transform and short-time autocorrelation, a new feature image that can characterize the signal is designed. This feature image has better anti-noise ability and the pixel features of the signal are more obvious at low SNR, as shown in Figure 4b. It can be seen that the short-time autocorrelation feature image in Figure 4b has a clearer pixel distribution than the original TFI in Figure 4a.

### 3.2. Signal Pre-Processing for Noise Reduction

Autocorrelation processing of the signal can effectively filters out the noise [21]. Given a signal x(t) and zero-mean Gaussian white noise w(t), the observable signal is expressed as y(t)=x(t)+w(t), and the autocorrelation function of y(t)
(10)$$\begin{array}{cc} R_{y}(\tau ) & =E[y(t)y(t-\tau )]=E\{[x(t)+w(t)][x(t-\tau )+w(t-\tau )]\}\\ & =E[x(t)x(t-\tau )]+E[w(t)w(t-\tau )]+E[x(t)w(t-\tau )]+E[w(t)x(t-\tau )]\\ & =R_{x}(\tau )+R_{w}(\tau )+R_{xw}(\tau )+R_{wx}(\tau ) \end{array}$$

We can conclude that $R_{xw}(\tau )=R_{wx}(\tau )=0$, and then $R_{y}(\tau )=R_{x}(\tau )+R_{w}(\tau )$ since the noise w(t) is not related to the signal x(t). For Zero-mean Gaussian white noise (ZGWN) w(t) with wide bandwidth, the autocorrelation function $R_{w}(\tau )$ mainly affects the nearby position τ=0, and when the value of τ is large, then $R_{x}(\tau )$ can be characterized approximately by $R_{y}(\tau )$, that is, $R_{y}(\tau )\approx R_{x}(\tau )$. For convenience, we write $R_{y}(\tau )$ as $R_{y}(t)$. Subsequently, combined with the TFT result of the autocorrelation function, $R_{y}(t)$ is updated according to the Choi-Williams distribution (CWD) [14]:
(11)$$C_{R}(t,\Omega )=\iint \sqrt{\frac{\pi \sigma }{\tau^{2}}}R_{y}(u+\tau /2)R_{y}{}^{*}(u-\tau /2)\times e^{-\pi^{2}\sigma {\left(u-t\right)}^{2}/4\tau^{2}-j\Omega \tau }dud\tau$$

According to Equation (11), the result that was obtained by the Choi-Williams transformation reflects the change of the autocorrelation function $R_{y}(t)$. As $R_{y}(t)$ is only determined by the signal x(t), the obtained time-frequency transform $C_{R}(t,\Omega )$ is a response of x(t) in the two-dimensional time-frequency plane after the autocorrelation domain, so it can represent the signal x(t) uniquely.

In the actual signal reception, the discrete value of y(t) represented by y(k) will be obtained after the IF sampling. In general, the autocorrelation function ${\hat{R}}_{y}(k)$ for discrete values of signals can be estimated while using the following formula
(12)$${\hat{R}}_{y}(k)=\sum_{n=0}^{N-k-1}y(n+k)y^{*}(n)$$
where, k=1,2,…,N−1 and N is the signal length. For the TFA of discrete signals, it is also necessary to use discrete Choi-Williams transform for calculation. x(k) is the value obtained by signal x(t) after IF sampling, and $T_{s}$ is the sampling interval of x(t). According to Equation (10), if $t=kT_{s}$, $\tau /2=mT_{s}$, $u=nT_{s}$, can get the equation, as follows
(13)$$C_{x}(k,\Omega )=2T_{s}^{2}\sum_{m}\sum_{n}\sqrt{\frac{\pi \sigma }{{\left(2mT_{s}\right)}^{2}}}x(nT_{s}+mT_{s})x^{*}(nT_{s}-mT_{s})\times e^{-\pi^{2}\sigma {\left(nT_{s}-kT_{s}\right)}^{2}/4{\left(2mT_{s}\right)}^{2}-j2\Omega mT_{s}}$$

Normalize $T_{s}$ to 1, and $\omega =\Omega T_{s}$, the result can be expressed, as follows
(14)$$C_{x}(k,\omega )=2\sum_{m}\sum_{n}\sqrt{\frac{\pi \sigma }{4m^{2}}}x(n+m)x^{*}(n-m)\times e^{-\pi^{2}\sigma {\left(n-k\right)}^{2}/16m^{2}-j2m\omega }$$

Assuming that the sampling length of x(k) is N, then the frequency ω can be discretized, and the discrete form of the Choi-Williams transform can be obtained, as follows.
(15)$$C_{x}(k,l)=2\sum_{m}\sum_{n}\sqrt{\frac{\pi \sigma }{4m^{2}}}x(n+m)x^{*}(n-m)\times e^{-\pi^{2}\sigma {\left(n-k\right)}^{2}/16m^{2}-j4\pi ml/N}$$

Taking ${\hat{R}}_{y}(k)$ into Equation (15), we can get two-dimensional features that are composed of signal autocorrelation sequences, as follows
(16)$$C_{R}(k,l)=2\sum_{m}\sum_{n}\sqrt{\frac{\pi \sigma }{4m^{2}}}{\hat{R}}_{y}(n+m){\hat{R}}_{y}{}^{*}(n-m)\;\times e^{-\pi^{2}\sigma {\left(n-k\right)}^{2}/16m^{2}-j4\pi ml/N}$$

Through the above analysis, it can be found that the signal autocorrelation has an inhibitory effect on noise. The two-dimensional feature $C_{R}(k,l)$ that is obtained by ${\hat{R}}_{y}(k)$ is the reflection of the discrete signal s(k) in the two-dimensional time-frequency plane after passing through the autocorrelation domain. Therefore, using $C_{R}(k,l)$ to classify the original signal can generate the feature image of the signal. In addition, when $C_{R}(k,l)$ is transformed into a two-dimensional image, $C_{R}(k,l)$ needs to be transformed into this pixel interval since the pixel value of the image ranges from 0 to 255. The autocorrelation feature image of the signal is shown in Figure 5.

The results of the signal autocorrelation reflect how relevant they are at any two different times. Therefore, the feature image that is generated by autocorrelation also reflects the corresponding information. The higher the pixel intensity of the feature image, the higher the degree of autocorrelation of the signal at this time. Although the feature images that are generated by autocorrelation can effectively reduce the effect of noise, this method is not applicable to signals with a low degree of autocorrelation. Owing to LPI radar signals, such as Costas, LFM, P1–P4, T1–T4, and Frank, which exhibit large changes over time in a pulse, are non-stationary signals, and have low autocorrelation. The feature images of the LFM, T1, and P1 signals, which are generated by autocorrelation, are shown in Figure 5. The pixels of the feature image are concentrated only in the middle of the image, and the effective pixel information is not included in the rest area of the image. As the middle region of the image represents the autocorrelation results without delay, and another region of the image represents the degree to which the signal is correlated at two different times, this further illustrates the effect of poor autocorrelation of such signals on feature images generated by autocorrelation. In this of view, this paper proposes a more universal signal classification method, aiming at the similarity of autocorrelation images of LPI radar signals.

The feature images of signals with poor autocorrelation, which are generated by autocorrelation, are too similar to be effectively classified. However, changes in the signal over a short period of time can still be seen as a smooth change, owing to the processing of non-stationary signals in the short-time Fourier transform, which usually uses a sliding window and Fourier transform on the sampling points in the window to analyze non-stationary signals. The time domain waveform of the LPI signal defined in this paper is shown in Figure 6.

The 1, 2, and 3 parts of these images are relatively stable waveform changes, which further demonstrates that the signal can be analyzed as a stationary signal in a short period of time. This paper constructs a feature image capable of characterizing the signal from the perspective of short-time autocorrelation, owing to the stationary signal having good correlation. In addition, using the short-time autocorrelation and the series of methods used in the construction of the autocorrelation feature image proposed in Section 3.3, the short-time autocorrelation feature image is shown in Figure 7. It can be found that the short-time autocorrelation images of the LFM, T1, and P1 signals have obvious pixel features that can characterize the signal. Therefore, for the signals with poor autocorrelation, the short-time autocorrelation feature images that are proposed in this paper can achieve signal classification.

### 3.3. Proposed Noise Reduction Algorithm for LPI Signal

In the short-time autocorrelation processing, given the IF sampling signal y(k) of length N, the signal is cut into several frames according to the window length M of the rectangular window, and M is also the length of one frame of the signal. L represents the number of short-time signals that formed by dividing y(k). That is, the number of frames of the signal. If the last frame length of the signal is less than M, then the frame signal will be padded with zeros. Figure 8 shows a specific schematic diagram of the signal segmentation process. In the process of segmentation according to the window length M, an overlap value W is also set, which indicates the length of the overlapping portions of the adjacent two frames. Finally, after segmentation, several frame signals can be obtained, which is represented as $y_{1}(k),y_{2}(k),\ldots ,y_{L}(k)$.

After the signals have being segmented, each frame of the signal is autocorrelated. Then, the autocorrelation value of each frame signal is intercepted to ensure that each frame signal retains a valid autocorrelation value while reducing the sample size. For the frame signal $y_{i}(k)$ with length M, an autocorrelation sequence ${\hat{R}}_{yi}(k)$ with 2M−1 can be obtained after autocorrelation, and then ${\hat{R}}_{yi}(k)$ will be clipped. We take (M/2,3M/2) of autocorrelation sequence ${\hat{R}}_{yi}(k)$ to form a new sequence ${\hat{R}}_{yi}{}^{\prime }(k)$. For convenience, function clip(•) is used to represent intercept methods. After interception, the sequence of each frame can be expressed as
(17)$${\hat{R}}_{yi}{}^{\prime }(k)=clip({\hat{R}}_{yi}(k))$$
where, i=1,2,…,L. Finally, the sequence of each frame obtained by short-time autocorrelation is spliced head-to-tail to obtain the short-time autocorrelation value of each phase of the intra-pulse signal, as follows
(18)$$\hat{R}{}^{\prime }(k)=splice({\hat{R}}_{y1}{}^{\prime }(k),{\hat{R}}_{y2}{}^{\prime }(k),\ldots ,{\hat{R}}_{yL}{}^{\prime }(k))$$
where, the function splice(•) represents a complete sequence capable of characterizing the signal, which consists of head-to-tail splicing of each frame autocorrelation values of the intercepted signal. It should be emphasized that this paper normalizes each frame after the signal is intercepted ensure that the autocorrelation values of each frame after the splicing can have the same weights since the autocorrelation results of the signals in each frame are different. That is, the spliced signal consists of a normalized autocorrelation value for each frame of the signal.

It is worth noting that, double short-time autocorrelation processing can be used on each frame sequence, which have been short-time autocorrelated, in order to enhance the classification effect of the classification network. Our previous research explained the calculation method of double and multiple autocorrelation [25]. For the specific implementation of double short-time autocorrelation, the input is also a signal, and then the signal is short-time autocorrelated. The difference between double short-time autocorrelation and short-time autocorrelation is that each autocorrelation sequence is autocorrelated again, and then the same clip(•) function and splice(•) function are used to realize the splicing of autocorrelation sequence. A second feature image that is capable of characterizing the original signal can be obtained in the same manner after obtaining the double short-time autocorrelation sequence of each frame signal. Thereby, two feature images representing the original signals can be obtained after the signal is subjected to short-time autocorrelation and double short-time autocorrelation. 

The autocorrelation image of the signal $y^{\prime }(k)$ is generated by combining Equations (7) and (16), as shown in Figure 9. In the simulation, M=20, W=5. Six non-stationary signals are shown in Figure 9, such as Frank, LFM, T1, T2, P1, and P2. When comparing Figure 9 with Figure 5, it can be found that, unlike the signal autocorrelation image, it is difficult to characterize the signal when the signal autocorrelation is poor. Correspondingly, the feature image that is generated by the short-time autocorrelation has more effective pixel features for the signal with poor autocorrelation. Therefore, short-time autocorrelation feature images can be used to classify the signals. At the same time, the non-stationary signal can be analyzed as a stationary signal in a short time. Just as the autocorrelation has the effect of noise reduction, the short-time autocorrelation also has this ability. Figure 10 shows the TFI and the short-time autocorrelation image of the LFM signal at SNR of 0 dB. It can be found that the pixel features of the TFI start to become blurred, while the short-time autocorrelation image still has obvious pixel features and it does not contain the pixel variation of the impurity, which further illustrates that the short-time autocorrelation image is robust under the effect of noise.

Since the sequence ${\hat{R}}_{yi}{}^{\prime }(k)$ that is generated by the short-time autocorrelation of each frame has a large difference, the Hamming window can more effectively analyze the sequence than the Kaiser window. Therefore, when constructing a short-time autocorrelation image, the Hamming window is selected as a window function for TFA. Figure 11 compares the short-time autocorrelation images generated by the LFM signals from the Hamming window and the Kaiser window, respectively. It can be found that the short-time autocorrelation feature image that is generated by the Kaiser window has unclear pixel features, which generated by the Hamming window has more stable and regular pixel features.

## 4. Design of the Proposed Hybrid Model

In this section, the Multiple Feature Images Joint Decision (MFIJD) model is designed to fully extract the pixel feature from the three feature images of the signals that are defined in Table 1. The three feature images are the original TFI, the short-time autocorrelation feature image, and the double short-time autocorrelation feature image, respectively. In MFIJD, the features are extracted by two structures. Structure 1 based on CNN, and structure 2 is composed of two depth models, CNN and BiLSTM. At the same time, an inference engine module based on fully connected structure is designed in MFIJD. The pre-classification results of the three feature images are processed by the inference engine module to effectively classify the LPI radar signal.

### 4.1. Design of Feature Extraction Structure

In [9,13,14,17], the initial input of the network is one TFI, while this paper proposes inputting three feature images represented by the same signal. Structure 1 is to extract pixel information of three feature images by using three sets of parallel CNN networks, as shown in Figure 12a. Structure 2 combines CNN and BiLSTM to extract features, and Figure 12b shows the structure. The short-time autocorrelation image $pic_{1}$ and the double short-time autocorrelation image $pic_{2}$ are, respectively, input into the corresponding CNN. The original TFI $pic_{3}$ is input into the BiLSTM for pixel feature extraction, and the internal structure of the BiLSTM extracted pixel feature is shown in Figure 13. According to the structural characteristics of BiLSTM, the TFI is divided into several sets of one-dimensional vectors that are input from head to tail and tail to head, respectively, to LSTM to achieve the effect of two-way long-term and short-time memory. The internal structure of the LSTM is composed of the cell unit cell, which is a one-way transmission. Then, each group of LSTM output results are input into the next group of LSTM, and the final pre-classification result is output from the fully connected layer. The dropout layer is added between the two layers of fully connected layers, and the probability of the dropout is set to 0.5 to prevent network overfitting.

In Figure 12, it is important to note that the three depth models output the pre-classification results, m represents the number of signal types to be classified in a given classification task. A total of 3m pre-classification results are obtained, which are input into the inference engine to obtain the classification result of the network, since the two feature extraction methods are both composed of three sets of depth models.

### 4.2. Network Parameter Adjustment

The parameter configuration schemes on the CNN network of Structure 1 and Structure 2 are the same and the networks in Structure 2 are connected in parallel, in order to effectively train the network model, it is necessary to unify the same parameters of BiLSTM and CNN. Therefore, the block size is set to 256 in BiLSTM and the learning rate is 0.01. For BiLSTM, the hyperparameters also include the number of forgotten gate offsets and hidden layer neurons. For the forgotten gate offset, setting it to 1 when the model is initialized can effectively avoid the output explosion or disappearance caused by the gradient at the beginning of the model. Therefore, the forgotten gate offset is set to 1. For the number of neurons in the hidden layer, the hyperparameters are changed to find a better parameter structure, and the performance of the network parameters are shown in Table 2. It should be emphasized that, the test set contains 11 types of signal samples from −6 dB to 9 dB. The training time and recognition rate are based on the GTX 1050Ti GPU and the simulation results obtained after 1000 iterations.

The results show that the recognition rate of the model is increasing when the number of neurons in the hidden layer of BiLSTM is less than 256 (numbers 1, 2, 3). The recognition rate of BiLSTM is 64.1% when the number of neurons is 256. When the number of neurons increases to 512, the training time consumption of the model is more than doubled, while the recognition rate is not improved. Therefore, the parameter configuration scheme of the BiLSTM model finally determined in this paper is: the block size is 256, learning rate is 0.01, the number of hidden layer neurons is 256, and the forgetting gate offset is 1.

## 5. Performance Demonstration and Comparison to the Competitive Literature

In this section, the paper evaluates the recognition performance of the MFIJD model on the test set. Firstly, the data set generation method of the simulation experiment is explained. Secondly, the proposed MFIJD model is compared with other radar signal waveform recognition methods. Finally, this paper further analyzes the recognition performance of MFIJD.

### 5.1. The Simulation Condition 

In the simulation, each parameter value of the signal is set to be randomly transformed within the specified range. The [a,b] represents that the corresponding parameter value takes any integer within the closed interval formed by a to b. In this paper, 11 types of radar signals that are defined in Table 1 are simulated, including LFM, Frank, Costas, P1–P4, and T1–T4. Table 3 shows the parameter ranges of these 11 types of signals. In the simulation, the signal sampling rate $f_{s}=50$ MHZ, U(a,b) represents that the signal parameters are evenly distributed within the specified range. $f_{\min}$ represents the greatest common divisor of each frequency of the Costas signal, $t_{p}$ represents the code period of the Costas signal, and $N_{c}$ represents the number of times that the Costas signal can be frequency switched.

The training set and test set are generated based on the parameter configuration of the 11 types of signals. The data sets are generated every 3 dB from −6 dB to 9 dB, while each type of signal produces 800 samples at each SNR, which are classified into a training set and a test set according to a ratio of 3:1. Therefore, the training set consists of 39,600 samples. The test set consists of 13,200 samples. The model is constructed while using Python’s tensorflow framework and the GPU is Nvidia 1050Ti.

### 5.2. Classification Result of Test Set

Figure 14 and Figure 15 show the change of the recognition rate and the loss value of the training set of Structure 1 and Structure 2 with the training times.

Figure 16 shows the overall recognition rate that was obtained by the proposed method in the test set. As the SNR increases, the recognition rate of the MFIJD model also increases. When the SNR is −9 dB, the recognition rates of Structure 1 is 60% and the recognition rates of Structure 2 is 50%, respectively. While the SNR is −6 dB, the recognition ratios of Structure 1 is 87% and the recognition ratios of Structure is 76%. When the SNR is greater than −3 dB, the MFIJD will have a recognition rate of more than 90% in the test set. However, on the whole, the recognition rate of structure 1 is higher than structure 2.

When the SNR is 0 dB, Figure 17 shows the confusion matrix of the signal. It can be found that comparing the two structures proposed in this paper, the classification results of 11 types of signals defined in Table 1 by structure 1 are less likely to be confused.

### 5.3. Comparison of Algorithm Performance

In this paper, the recognition performance of the MFIJD model is compared with the methods that were proposed in [13,15,16]. In [13], eight types of signals were used for simulation analysis, including LFM, Costas, BPSK, Frank, and T1–T4. In [16], it also simulated eight types of signals, unlike [13], the simulation uses P1–P4 instead of T1–T4. In [15], the simulation analyzed 12 types of signals of LFM, Costas, BPSK, Frank, P1–P4, and T1–T4. The signal classification techniques used in [13,15,16] focus on different signals, respectively. Therefore, by comparing the above literatures, the classification models that were proposed in this paper can be comprehensively compared.

In [13], Zhang proposed combining image morphology with threshold filtering to remove a large amount of noise from the image while using threshold denoising after the signal is transformed to a TFI, then the image morphology is used to remove the weak noise of the TFI. In order to compare the model of this paper with [13], this paper uses the same simulation conditions as [13], the SNR range is −3 dB to 9 dB, while 1000 test samples for each signal at each SNR. Figure 18 shows a comparison of the classification results of this paper with the literature [13] on the test set. The results show that the MFIJD model that is based on Structure 1 and Structure 2 has better recognition performance than the CNN model of [13], which mainly reflected in the recognition rate and recognition stability. For the convenience of expression, we refer to the network structure proposed in [13] as CNN. For structure 1, when the SNR is −3 dB, except for the T2 and T3 signals, the classification result of structure 1 is better than CNN that is proposed in [13]. Nonetheless, CNN has no significant improvement in the recognition rate of these two types of signals when compared to Structure 1. The recognition rates of structure 1 for T2 and T3 are 98% and 93%, while the recognition rates of CNN for T2 and T3 are 100% and 95%, respectively. When comparing the recognition rates of the other six types of signals, the recognition rates of Structure 1 for Frank, T1, T4, LFM, Costas and BPSK are 18%, 0%, 23%, 18%, 3%, and 6% higher than CNN, respectively. When the signal-to-noise ratio is greater than −3 dB, the recognition rate of structure 1 for each type of signal is not less than 99%. The recognition rate of Structure 2 for T1, T2, and T3 signals are not higher than CNN; however, when the SNR is increased, only the recognition rates of T1 and T3 in structure 2 are slightly lower than CNN. For recognition stability, it is difficult for CNN to find an exact trend of recognition rate change. Specifically, the recognition rate curve of CNN for various types of signals varies greatly with the different SNR. For CNN in [15], BPSK, Costas, T1, T2, and T3 are signals with high recognition rate, LFM, Frank, and T4 are signals with low recognition rate. For the three types of signals with low recognition rate, the recognition rate curve of LFM changes gently, Frank’s recognition rate curve changes steeply, and the recognition rate curve of T4 changes greatly. While the MFIJD has a relatively consistent trend in the recognition rate curves of various types of signals. As the SNR increases, the recognition rate of MFIJD rises faster at low SNR, which changes greatly at high SNR. Therefore, MFIJD has higher stability.

In [16], Liu proposes extracting signal characteristics from the power spectral density, moment accumulation, time-frequency distribution and transient characteristics of the signal, and proposes three new signal characteristics, which then use Artificial Bee Colony (ABC) algorithm optimizes the SVM classifier and, finally, uses SVM to classify the extracted signal features. When compared with [16], in the same simulation conditions, the SNR range is from −6 dB to 9 dB, and each test signal generates 1000 test samples. Figure 19 compares with the classification results of literature [16] on the test set.

For the recognition rate of [16], the structure 1 of the MFIJD model is similar to ABC-SVM, while the Structure 2 is not as effective as ABC-SVM. Figure 19 shows the overall recognition rate. For structure 1, when the SNR is from −6 to −3 dB, the overall recognition rate is slightly higher than ABC-SVM. While the SNR is from −3 to 3 dB, the overall recognition rate is slightly lower than ABC-SVM. At the same time, for the classification results of LFM, Costas, BPSK, P2 and P4, Structure 1, and ABC-SVM are almost the same, while for the classification results of Frank, P1 and P3, Structure 1, and ABC-SVM are different. When the SNR is −6 dB, the recognition rates of ABC-SVM for the three signals are 75%, 71%, and 72%, which of Structure 1 are 98%, 51%, and 90%, respectively. That is to say, the overall recognition performance of Structure 1 and ABC-SVM is comparable, but there are still differences in the classification effect of a few signals. For structure 2, the classification results for BPSK and P2 are comparable to ABC-SVM. For Frank and P3, the recognition rate is higher than ABC-SVM. However, for LFM, Costas, P1, and P4, the recognition rate of structure 2 is lower than those of ABC-SVM.

In [15], Zhang et al extracts the time domain features and time-frequency domain features of the signal. In the time domain, they extract signal features based on power spectral density, statistics, and transient characteristics. In the time-frequency domain, they extract signal image features that are based on binarized images. In summary, Zhang et al. proposed using CNN and ENN to design a hybrid classifier to classify signals base on the extracted feature. For the convenience of description, the hybrid classifier combining CNN and ENN in [15] is represented by NN. When compared with [15], in the same simulation conditions, the SNR range is from −6 dB to 9 dB, and each test signal generates 1000 test samples. Figure 20 compares with the classification results of literature [15] on the test set.

When compared with the recognition rate of [15], it can be seen that Structure 1 and Structure 2 are significantly better than NN, and their overall recognition rates are shown in Figure 20 (m). Specifically, for structure 1, the recognition rate of NN for P1, T1, and T2 is slightly higher than structure 1 at certain SNR. In addition, the recognition rate of structure 2 for the other signals is higher than NN. For Frank, LFM, P1–P4, the recognition rate of structure 1 at low SNR is much higher than NN, so, when classifying these 12 types of radar signals, Structure 1 has a greater advantage than NN. For Structure 2, the recognition rate for Frank, LFM, Costas, BPSK, and P2–P4 is higher than NN. Note that for the LFM, Costas, and BPSK, the recognition curves of Structure 1 and Structure 2 are basically coincident, while the classification results for P1 and T1–T4, Structure 2 are lower than NN. The recognition rate of Structure 2 for Frank, LFM, and P2–P4 is 17%, 25%, 25%, 27%, and 10% higher than NN, respectively, at SNR of −3 dB. While for NN, the recognition rate for P1 and T1–T3 is 7%, 18%, 6%, and 9% higher than Structure 2. Therefore, the conclusion is that the recognition performance of structure 2 is better than NN based on overall recognition.

While comparing the performance of the algorithms, we also compared three LPI radar signal classification techniques. When compared with [16], the recognition performance of structure 1 proposed in this paper is equivalent. When compared with [13,15], the two structures proposed in this paper have better recognition performance. For the classification of the 12 types of radar signals defined in this paper, the technique proposed in this paper has a high recognition rate and higher universality. Therefore, this technique can be used to classify non-stationary signals with poor autocorrelation. However, there are still some problems in the simulation that have not been analyzed. In the signal preprocessing proposed in this paper, three feature images that are capable of characterizing the signal are obtained in this paper. Therefore, we further analyze the effect of the number of feature images on the classification results in Section 5.4. It can be found that the overall performance of the structure 1 is better than that of the structure 2. The only difference between Structure 1 and Structure 2 is the depth model built during the feature extraction. Therefore, we also further analyze the effect of different depth models on the classification results in Section 5.4.

### 5.4. MFIJD Model Analysis 

For the MFIJD model, we analyze the effect of the number of feature images on the classification results by changing the input feature images. A total of five cases are analyzed, and Table 4 shows the input feature image and the corresponding feature extraction model. It should be noted that the multiple short-time autocorrelation feature image in Table 4 is two feature images that are generated by short-time autocorrelation and double short-time autocorrelation combined with TFA. The NLMS TFI is generated by the Normalized Least Mean Square (NLMS) algorithm adaptive filtering combined with TFA. The feature extraction model used in Table 4 corresponds to the feature image type. For example, the input feature image type is a short-time autocorrelation feature map + NLMS TFI, which indicates that the feature extraction model is CNN and BiLSTM. This means that CNN extracts the pixel features of the short-time autocorrelation feature image, and BiLSTM extracts the pixel features of the NLMS TFI.

As shown in Figure 21, it is found that the signal classification result is greatly improved when two feature images are input. Specifically, when the SNR is −9 dB, if a single image is input, then the recognition rate of the model is low. For example, the short-time autocorrelation feature image and the NLMS TFI are input to CNN and BiLSTM, respectively, and the recognition rate is only 41% and 25% at a low SNR of −9 dB. However, if the number of input feature images is increased, the recognition rate of the model is increased by at least 10%. It is worth noting that when the multiple short-time autocorrelation feature image (STAFI) and the NLMS time-frequency image are input, the recognition rate of the method is not the highest when the SNR is less than −6 dB. While the recognition rate is only higher when the SNR is greater than −6 dB. This means that simply increasing the number of input feature images could not significantly improve the classification results for the 12 LPI radar signals defined in this paper.

Some depth models are used to classify signals, however, it can be found from the Section 5.4 that the recognition rate of three sets of parallel CNN networks is better than the two sets of CNNs combined with BiLSTM, so we further analyze the effect of different models. It should be noted that the feature extraction model combinations used in Table 5 correspond to short-time autocorrelation images, double short-time autocorrelation images, and NLMS TFI, respectively. For example, the feature extraction model is combined with CNN+CNN+BiLSTM, which uses the CNN to extract the pixel features from the short-time autocorrelation feature image, uses the CNN to extract the pixel features from the double short-time autocorrelation feature image, and then uses the BiLSTM to extract the NLMS TFI. The effect of different feature extraction models on the recognition rate based on Table 5 is shown in Figure 22 in order to facilitate the intuitive analysis.

It can be found that CNN can more effectively extract the pixel features of the feature image. When used BiLSTM, the recognition rate of the model is the lowest, and the recognition rate is 43.2% at the SNR of −9 dB. While used CNN, the recognition rate is the highest, and recognition rate is 59.3% at the SNR of −9 dB. In addition, for the feature extraction model contains two sets of CNNs, the recognition rate is higher than the model containing two sets of BiLSTM. Therefore, three parallel CNN networks can be used to obtain better classification results for the short-time autocorrelation feature image and the NLMS TFI.

## 6. Conclusions

The MFIJD model that is proposed in this paper provides more initial information for the network by inputting multiple feature images, which is suitable for non-stationary signals and has higher universality. It is found the simultaneous input of multiple feature images into a parallel depth model can significantly improves the classification results by analyzing the effect of multiple feature images in MFIJD on the signal recognition rate. For the two depth models, CNN has better signal classification results than BiLSTM. Therefore, the model using three parallel CNN networks is more suitable for processing short-time autocorrelation feature images proposed in this paper to extract pixel features. The comparison with the literature shows that the MFIJD model that was proposed in this paper can classify 11 types of LPI radar signals and has better recognition performance. The model not only has a high recognition rate at high SNR, but also better waveform classification in a low SNR. Finally, we further analyze the effects of MFIJD on the waveform recognition rate, time consumption and image similarity of LPI radar signals, and verifies the rationality of the network structure. From the accuracy of recognition rate and recognition stability, the MFIJD model achieves better recognition performance.

## Figures and Tables

**Figure 1 sensors-20-00526-f001:**
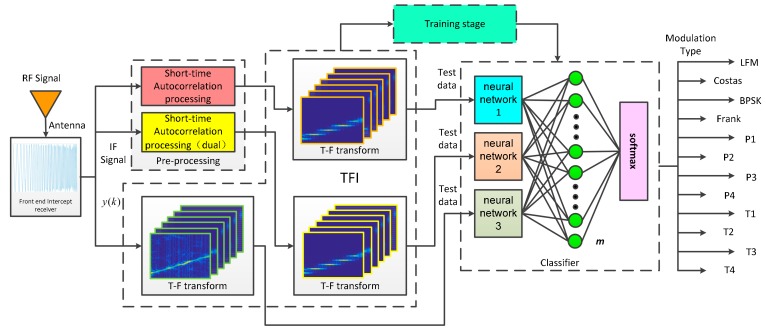
The framework of proposed LPI radar waveform recognition technique (LWRT).

**Figure 2 sensors-20-00526-f002:**
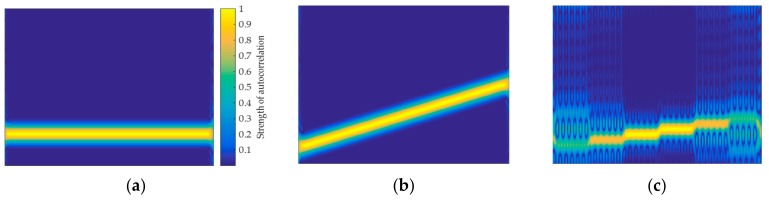
Time-frequency feature images of signals: (**a**) Single frequency; (**b**) linear frequency modulation (LFM); (**c**) P1.

**Figure 3 sensors-20-00526-f003:**
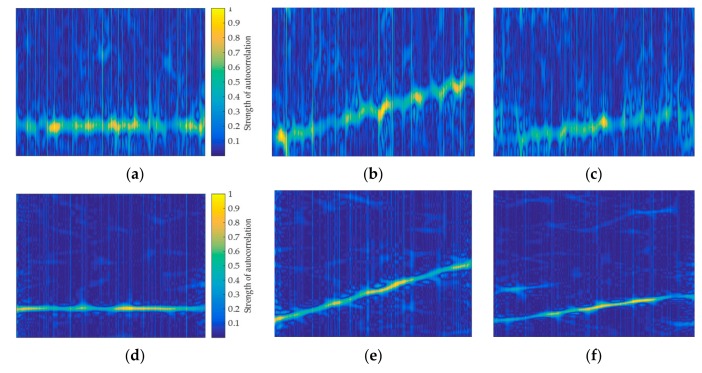
The signal time-frequency images (TFIs) when the SNR is 0dB, the first row represent the images generated by Kaiser window, and the second row represent the images generated by Hamming window: (**a**) Single frequency; (**b**) LFM; (**c**) P1; (**d**) Single frequency; (**e**) LFM; and, (**f**) P1.

**Figure 4 sensors-20-00526-f004:**
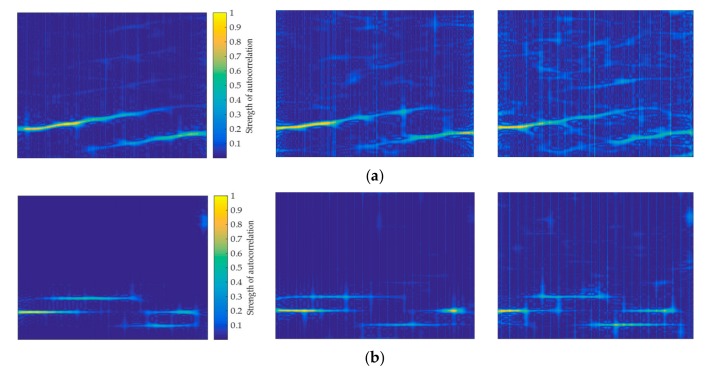
Feature images formed from left to right when SNR is 3 dB, 0 dB, and −3 dB, respectively: (**a**) Unprocessed signal TFI of Frank; and, (**b**) Short-time autocorrelation feature Image of Frank.

**Figure 5 sensors-20-00526-f005:**
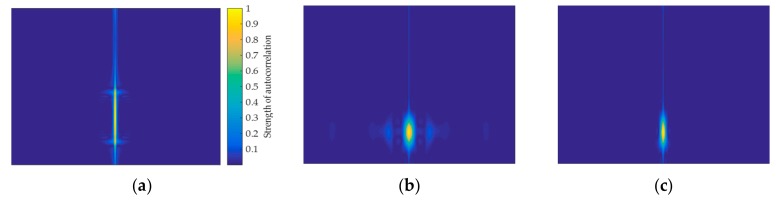
Autocorrelation feature images formed by LFM, T1, and P1 signal: (**a**) LFM; (**b**) T1; and, (**c**) P1.

**Figure 6 sensors-20-00526-f006:**
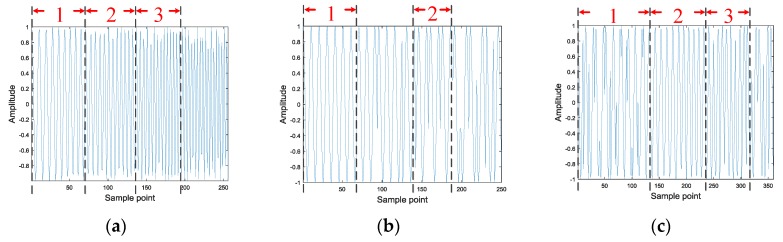
Time domain waveform analysis of LFM, T1 and P1: (**a**) LFM; (**b**) T1; and, (**c**) P1.

**Figure 7 sensors-20-00526-f007:**
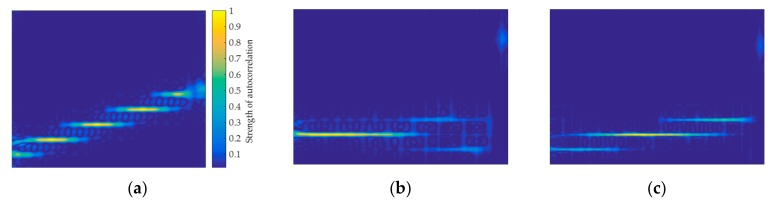
Short-time autocorrelation images of LFM, T1, and P1 signals: (**a**) LFM; (**b**) T1; and, (**c**) P1.

**Figure 8 sensors-20-00526-f008:**
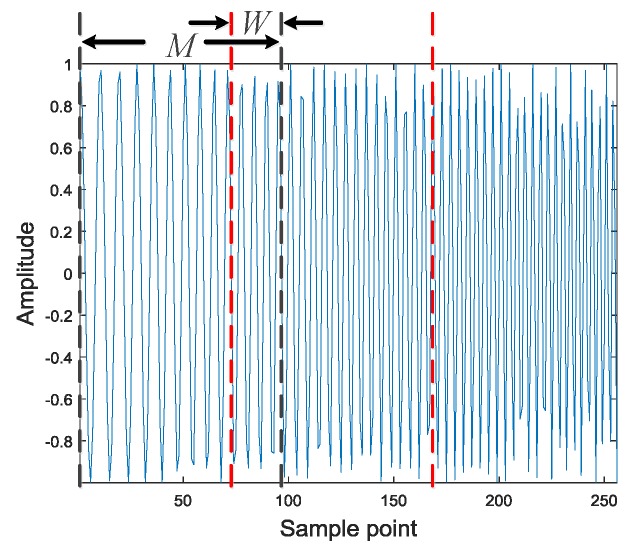
Signal segmentation diagram.

**Figure 9 sensors-20-00526-f009:**
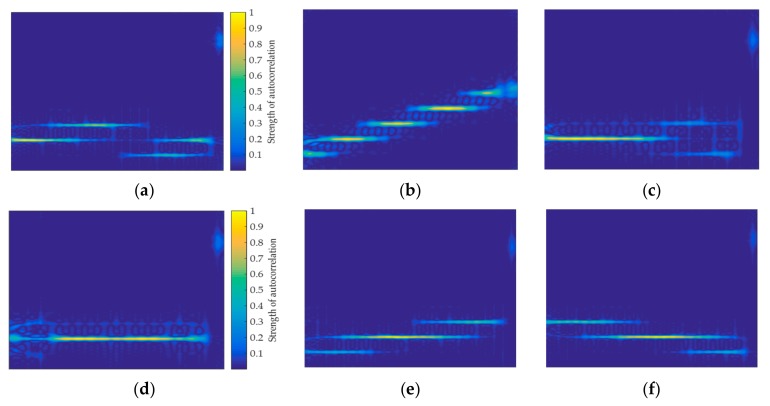
Short-time autocorrelation images of six signals: (**a**) Frank; (**b**) LFM; (**c**) T1; (**d**) T2; (**e**) P1; and, (**f**) P2.

**Figure 10 sensors-20-00526-f010:**
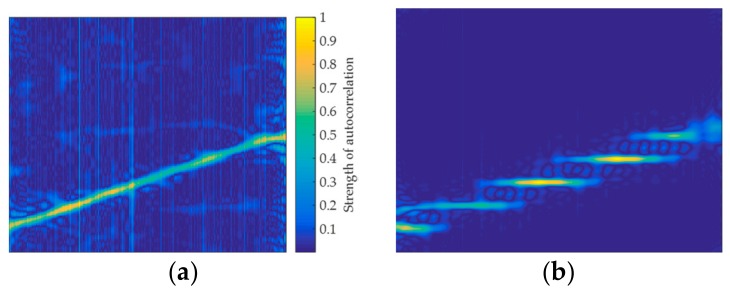
Noise reduction ability analysis of short-time auto-correlation image when SNR is 0 dB: (**a**) Time-frequency image of LFM. (**b**) Autocorrelation image of LFM.

**Figure 11 sensors-20-00526-f011:**
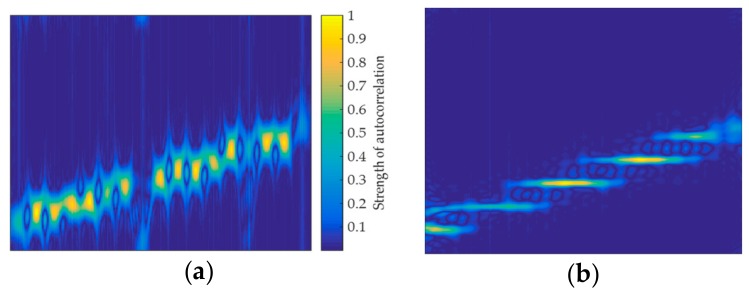
Window function selection analysis of short-time auto-correlation images when SNR is 0 dB: (**a**) Kaiser window for LFM. (**b**) Hamming window for LFM.

**Figure 12 sensors-20-00526-f012:**
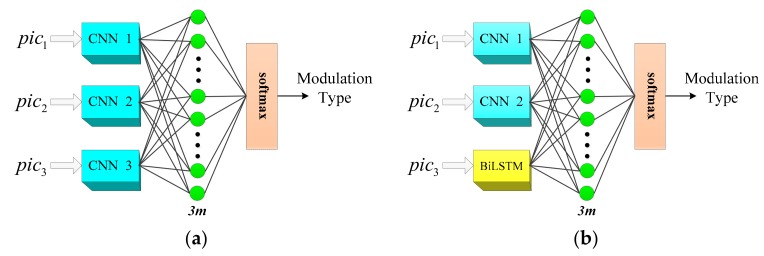
The overall structure framework of the multiple feature images joint decision (MFIJD): (**a**) Structure1. (**b**) Structure2.

**Figure 13 sensors-20-00526-f013:**
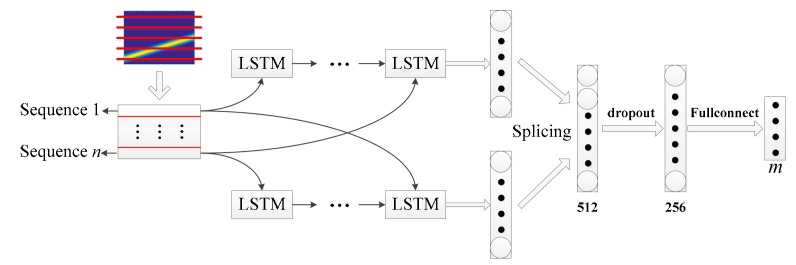
Internal structure of BiLSTM.

**Figure 14 sensors-20-00526-f014:**
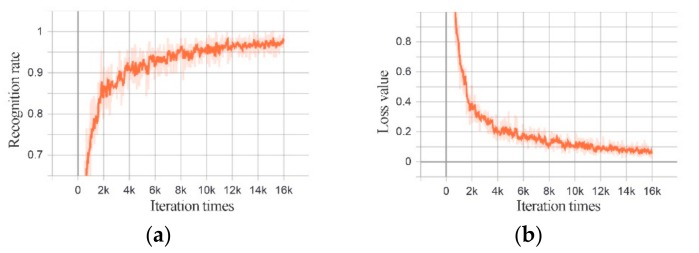
MFIJD model structure 1 training process: (**a**) The change of recognition rate of training set with the number of iterations. (**b**) The change of loss function with the number of iterations.

**Figure 15 sensors-20-00526-f015:**
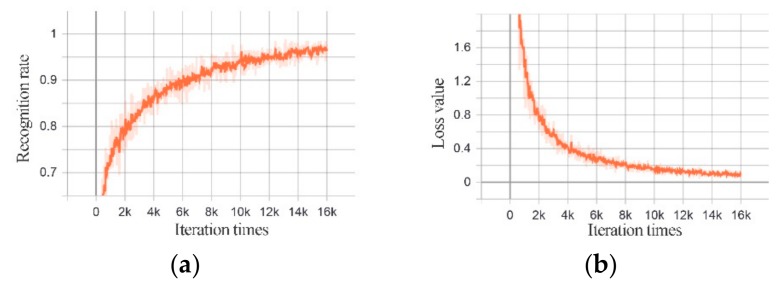
MFIJD model structure 2 training process. (**a**) The change of recognition rate of training set with the number of iterations. (**b**) The change of loss function with the number of iterations.

**Figure 16 sensors-20-00526-f016:**
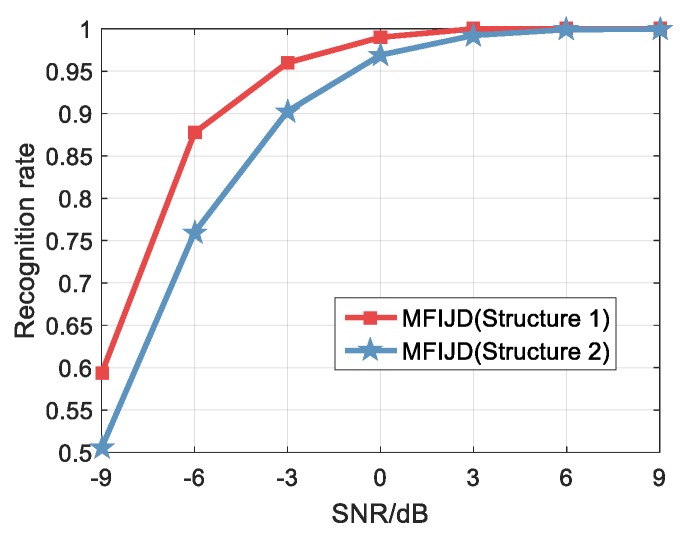
The classification result of the MFIJD in the test set.

**Figure 17 sensors-20-00526-f017:**
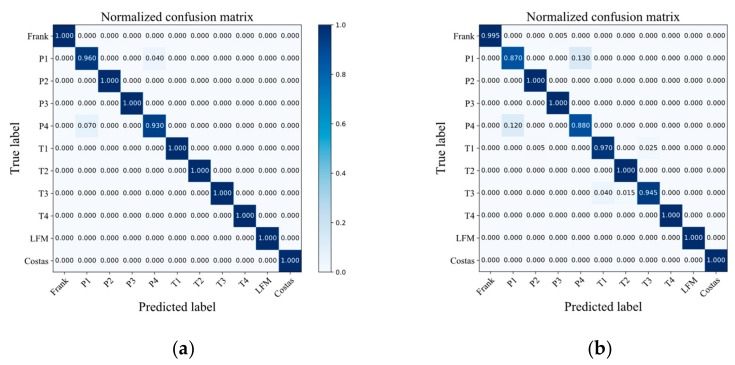
Confusion Matrix of MFIJD on 0 dB Test Set: (**a**) Structure 1. (**b**) Structure 2.

**Figure 18 sensors-20-00526-f018:**
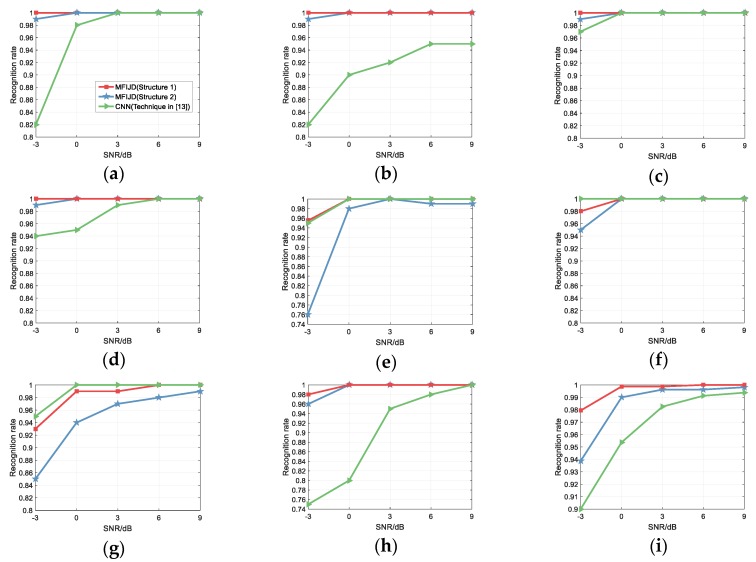
Comparison with the waveform classification in [13]: (**a**) Frank; (**b**) LFM; (**c**) Costas; (**d**) BPSK; (**e**) T1; (**f**) T2; (**g**) T3; (**h**) T4; and, (**i**) Overall.

**Figure 19 sensors-20-00526-f019:**
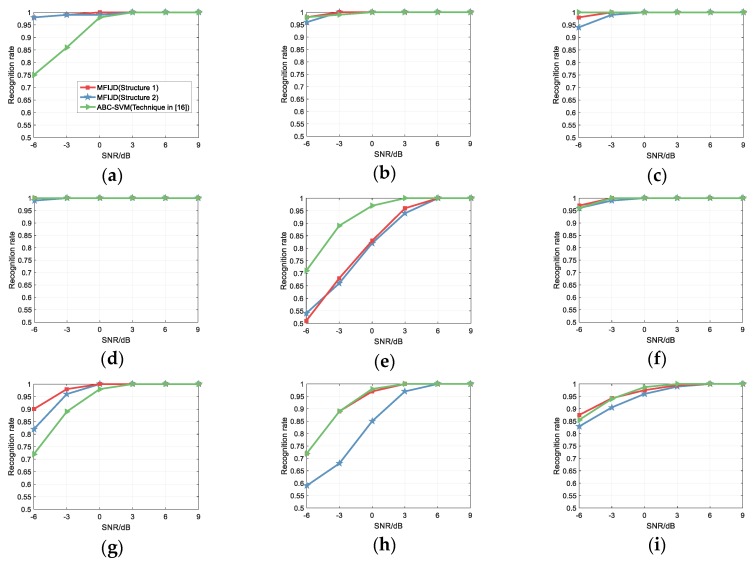
Comparison with the waveform classification in [16]: (**a**) Frank; (**b**) LFM; (**c**) Costas; (**d**) BPSK; (**e**) P1; (**f**) P2; (**g**) P3; (**h**) P4; and, (**i**) Overall.

**Figure 20 sensors-20-00526-f020:**
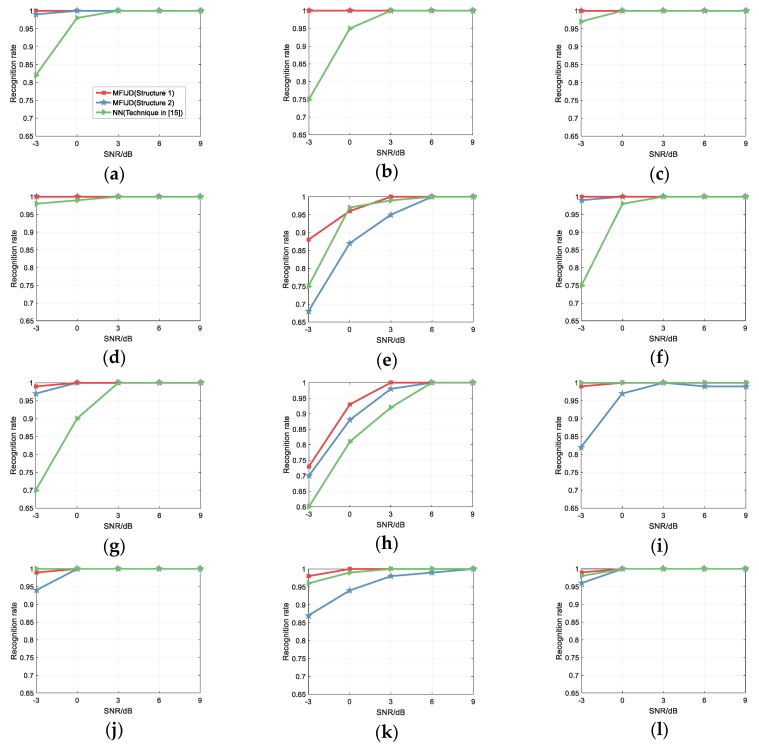
Comparison with the waveform classification in [15]: (**a**) Frank; (**b**) LFM; (**c**) Costas; (**d**) BPSK; (**e**) P1; (**f**) P2; (**g**) P3; (**h**) P4; (**i**) T1; (**j**) T2; (**k**) T3; (**l**) T4; and, (**m**) Overall.

**Figure 21 sensors-20-00526-f021:**
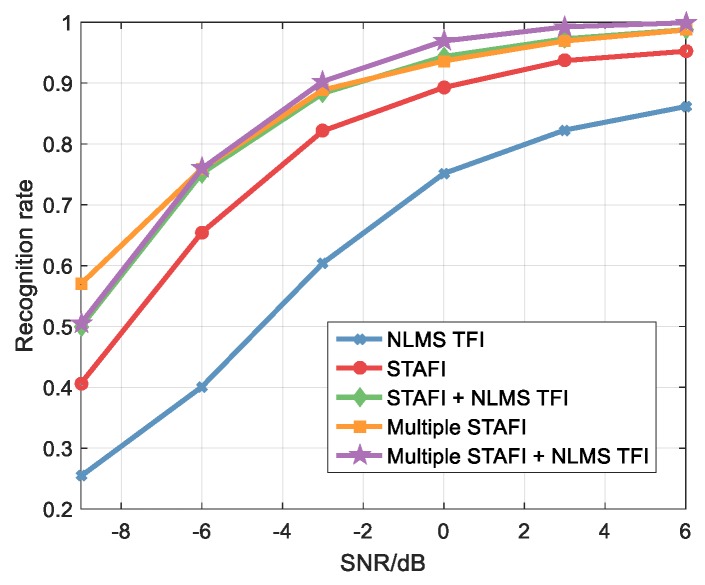
Recognition results according to different feature images.

**Figure 22 sensors-20-00526-f022:**
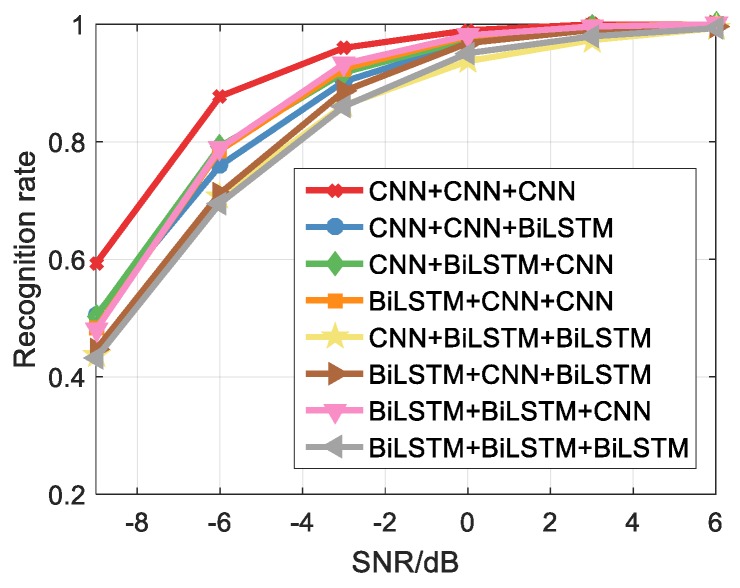
The effect of different feature extraction models on the recognition rate.

**Table 1 sensors-20-00526-t001:** Phase modulation signal parameter.

Modulation Type	f(k)	ϕ(k)
LFM	$f_{0}+\frac{B(kT_{s})}{\tau_{pw}}$	constant
Costas	*f* _j_	constant
Frank	constant	$\frac{2\pi }{M}(i-1)(j-1)$
P1	constant	$-\frac{\pi }{M}[(M-(2j-1))][(j-1)M+(i-1)]$
P2	constant	$-\frac{\pi }{2M}(2i-1-M)(2j-1-M)$
P3	constant	$\frac{\pi }{\rho }{\left(i-1\right)}^{2}$
P4	constant	$\frac{\pi }{\rho }{\left(i-1\right)}^{2}-\pi (i-1)$
T1	constant	$\frac{2\pi }{N_{ps}}\left\lfloor \frac{N_{ps}}{2\pi }\mathrm{mod}\{\frac{2\pi }{N_{ps}}\left\lfloor (N_{si}(kT_{s})-j\tau_{pw})\frac{jN_{ps}}{\tau_{pw}}\right\rfloor ,2\pi \}\right\rfloor$
T2	constant	$\frac{2\pi }{N_{ps}}\left\lfloor \frac{N_{ps}}{2\pi }\mathrm{mod}\{\frac{2\pi }{N_{ps}}\left\lfloor (N_{si}(kT_{s})-j\tau_{pw})(\frac{2j-N_{si}+1}{\tau_{pw}})\frac{N_{ps}}{2}\right\rfloor ,2\pi \}\right\rfloor$
T3	constant	$\frac{2\pi }{N_{ps}}\left\lfloor \frac{N_{ps}}{2\pi }\mathrm{mod}\{\frac{2\pi }{N_{ps}}\left\lfloor \frac{N_{ps}B{\left(kT_{s}\right)}^{2}}{2\tau_{pw}}\right\rfloor ,2\pi \}\right\rfloor$
T4	constant	$\frac{2\pi }{N_{ps}}\left\lfloor \frac{N_{ps}}{2\pi }\mathrm{mod}\{\frac{2\pi }{N_{ps}}\left\lfloor \frac{N_{ps}B{\left(kT_{s}\right)}^{2}}{2\tau_{pw}}-\frac{N_{ps}f_{c}(kT_{s})}{2}\right\rfloor ,2\pi \}\right\rfloor$

**Table 2 sensors-20-00526-t002:** Hyper parametric tuning of the BiLSTM Model.

Serial Number	Number of Neurons in the Hidden Layer	Time Consumption (min)	Recognition Rate (%)
1	64	2	50.8
2	128	4	55.9
3	256	7	64.1
4	512	12	64.1

**Table 3 sensors-20-00526-t003:** The range of signal parameter values.

Radar Waveforms	Parameters	Value of Range
LFM	$f_{c}$ $\tau_{pw}$ B	$U(f_{s}/6,f_{s}/4)$ $U(5\times 10^{-6},7.5\times 10^{-6})$ $U(f_{s}/20,f_{s}/10)$
Costas	$f_{\min}$ $t_{p}$ $N_{c}$	$U(f_{s}/30,f_{s}/20)$ $U(5\times 10^{-7},10^{-6})$ [3,6]
Frank, P1	$f_{c}$ M	$U(f_{s}/6,f_{s}/4)$ [6,8]
P2	$f_{c}$ M	$U(f_{s}/6,f_{s}/4)$ {6,8,10}
P3, P4	$f_{c}$ ρ	$U(f_{s}/6,f_{s}/4)$ {36,49,64}
T1, T2	$f_{c}$ $\tau_{pw}$ $N_{ps}$ $N_{si}$	$U(f_{s}/6,f_{s}/4)$ $U(5\times 10^{-6},7.5\times 10^{-6})$ [2,4] [4,6]
T3, T4	$f_{c}$ B $\tau_{pw}$ $N_{ps}$	$U(f_{s}/6,f_{s}/4)$ $U(f_{s}/20,f_{s}/10)$ $U(5\times 10^{-6},7.5\times 10^{-6})$ [2,4]

**Table 4 sensors-20-00526-t004:** Different feature images and feature extraction models.

Type of Feature Image	Feature Extraction Model
NLMS TFI	BiLSTM
Short-time autocorrelation feature image	CNN
Short-time autocorrelation feature image + NLMS TFI	CNN+BiLSTM
Multiple short-time autocorrelation feature image	CNN+CNN
Multiple short-time autocorrelation feature image + NLMS TFI	CNN+CNN+BiLSTM

**Table 5 sensors-20-00526-t005:** The effect of different feature extraction models on the recognition rate.

Combination of Feature Extraction Models	−9 dB	−6 dB	−3 dB	0 dB	3 dB	6 dB	9 dB
BiLSTM+BiLSTM+BiLSTM	0.432	0.695	0.860	0.951	0.980	0.994	0.995
BiLSTM+BiLSTM+CNN	0.480	0.789	0.933	0.981	0.996	1	1
BiLSTM+CNN+BiLSTM	0.446	0.712	0.886	0.968	0.990	0.997	0.999
CNN+BiLSTM+BiLSTM	0.437	0.708	0.862	0.937	0.973	0.994	0.995
CNN+CNN+BiLSTM	0.505	0.760	0.902	0.969	0.992	0.999	0.999
BiLSTM+CNN+CNN	0.482	0.785	0.925	0.980	0.997	1	1
CNN+BiLSTM+CNN	0.502	0.792	0.918	0.974	0.996	1	1

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
