# Peer review of "LPI Radar Waveform Recognition Based on Features from Multiple Images"

_sensors, 2020, doi:10.3390/s20020526_

Round 1

Reviewer 1 Report

This paper designed a new feature image which can characterize the non-stationary signals in a low SNR based on short-time autocorrelation, and analyzed the performance of MFIJD, which is of interest to the LPI radar waveform recognition community. However, I have a few suggestions before accepting this paper for publication.

1.In abstract,the abbreviation 'TFI' should be declared.
2.Below Eq.(5), the sentence "the noise $s(t)$" should be correctted.
3.Below Eq.(6), the signal $x(t)$ isn't explained,which is confused with the signal $s(t)$.
4.The paper writing must be polished.

Author Response

Point 1:  1.In abstract, the abbreviation 'TFI' should be declared.

Response 1: Thanks for the reviewer's reminder, this is indeed a mistake in the manuscript. We have modified according to reviewer's comments. ( line 18 )

Point 2: Below Eq.(5), the sentence "the noise s(t)" should be correctted.

Response 2: Thanks for the reviewer's reminder, this is indeed a mistake in the manuscript. We have modified according to reviewer's comments. ( lines 212 - 218 )

Point 3: Below Eq.(6), the signal x(t) isn't explained, which is confused with the signal s(t).

Response 3: Thanks for the reviewer's reminder, this is indeed a mistake in the manuscript. We have modified according to reviewer's comments. ( lines 221 - 224)

Point 4: The paper writing must be polished.

Response 4: Thanks for the reviewer's reminder, we have modified according to reviewer's comments, and some parts have been rewritten.( lines 42 - 48, lines 52 - 56, lines 62 - 92, lines 148 - 174, lines 290 - 296, lines 306 - 317, lines 333- 339 , lines 582 - 595 ). The paper has been polished.

Reviewer 2 Report

The authors proposed an algorithm to classify radar waveform by using time-frequency-analysis. The topic is interesting, and the results (at least with simulation data) are very promising. 

However, the paper cannot convince me in its current form. It seem that this manuscript was not seriously handled, and the consistency among sections is not maintained. For instance:

the variable notations in line 296, 301, 321, 326 and 341 are different from those on page 8. the paragraph between line 302 and 308 is duplicated to the content between 315 and 320.

Generally, I have several overall concerns / comments to the theory part:

the introduction of image processing method since line 68 is very chaotic. Normally, the existing approaches/algorithms should be arranged and introduced either according to their categories or their advantage and disadvantages. Some underlying logic is used to organize the description and introduction.  Some methods are just mentioned with their names without any briefing or explanation on what they are. It helps the reader nothing to have the overview of previous work. There are many time frequency analysis in the literature. Why do the authors choose the CWD. It is not a new method. The wavelet and its variants were intensively discussed in the last two decades for time-frequency analysis. Recently, the tensor structure based method has also been proposed, cf. "Enhanced Cohen class time–frequency methods based on a structure tensor analysis: Applications to ISAR processing" of Corretja et al. More recently, the neural network has also been adopted, cf. "Enhanced Cohen class time–frequency methods based on a structure tensor analysis: Applications to ISAR processing" of Bertoluzzo et al.  What is the motivation double short-time auto-correlation? And the double short time auto-correlation is not clearly explained, either. What is the input and what is the output ? Two structures are given in Figure 13. How can we fit these two structures in the framework in Fig. 1 ?  What is the purpose of the path "Training data pre-training" in Fig. 1?  Two window functions are mentioned and compared. Which one is then finally applied in the algorithm ?

Some detailed comments:

Line 85, a right bracket is redundant.  What does TFI stand for ? The notations used in Table 1 are not defined before using Line 167, equation (2). I assume that the phase should be generally the integral of a time varied frequency function over the time interval. In Figure 2, what does CP stand for ? Regarding plots in Fig. 3, why do the plots with worse SNR (-9 dB) in the second row seem to have a better contrast than those in the first row with SNR = 0 dB ?  Line 340, Equaiton (18) does not exist. 

Author Response

Response to Reviewer 2 Comments

Point 1: The variable notations in line 296, 301, 321, 326 and 341 are different from those on page 8.

Response 1: Thanks for the reviewer's reminder, this is indeed a mistake in the manuscript. We have unified the variable notations. ( line 282, 287, 295, 297, 318 )

Point 2: The paragraph between line 302 and 308 is duplicated to the content between 315 and 320.

Response 2: Thanks for the reviewer's reminder, this is indeed a mistake in the manuscript. We have modified according to reviewer's comments. ( lines 290 - 296 )

Point 3: The introduction of image processing method since line 68 is very chaotic. Normally, the existing approaches/algorithms should be arranged and introduced either according to their categories or their advantage and disadvantages. Some underlying logic is used to organize the description and introduction.

Response 3: Thanks for the reviewer's reminder, we have modified according to reviewer's comments and rewritten the introduction. ( lines 62 - 92 )

Point 4: Some methods are just mentioned with their names without any briefing or explanation on what they are. It helps the reader nothing to have the overview of previous work.

Response 4: Thanks for the reviewer's reminder, we have modified according to reviewer's comments and add the explanation of CWD for TFI ( lines 148 - 174 ), double short-time autocorrelation ( lines 306 - 317 ),  choice of window function ( lines 333 - 339 ).

Point 5: There are many time frequency analysis in the literature. Why do the authors choose the CWD. It is not a new method. The wavelet and its variants were intensively discussed in the last two decades for time-frequency analysis. Recently, the tensor structure based method has also been proposed, cf. "Enhanced Cohen class time–frequency methods based on a structure tensor analysis: Applications to ISAR processing" of Corretja et al. More recently, the neural network has also been adopted, cf. "Enhanced Cohen class time–frequency methods based on a structure tensor analysis: Applications to ISAR processing" of Bertoluzzo et al.  

Response 5: Thanks for the reviewer's questions,this is indeed a oversight in the manuscript. This manuscript does not describe the research problem clearly. As a modification, we have made a more detailed explanation in the Section 3.1. The research content of this manuscript is the classification of LPI radar signal. Specifically, when the receiver receives the radar signal, it first carries out signal sorting, and the sorted signal will use the classification method of this manuscript to determine the possible modulation type of the signal. Similar to the previous literature [6], [13], [14], [15] and [16], this manuscript also focuses on the classification of signal modulation methods. Therefore, it is envisaged that each set of signal data obtained after signal sorting represents a specific radar signal. In this research background, the results of bilinear time-frequency analysis do not produce cross term interference, so this manuscript does not do more analysis on the selection of time-frequency analysis methods. In the book Detecting and Classifying Low Probability of Intercept Radar written by Phillip E. Pace, the author performed a time-frequency analysis on LPI signals, which focused on the impact of CWD on LPI signals. According to the analysis of the book, we find that each LPI radar waveform shows a distinguishable mode after passing through CWD. Therefore, based on its analysis, this manuscript also chooses CWD to realize time-frequency transform for the signal. ( lines 148 - 174 )

Point 6: What is the motivation double short-time auto-correlation? And the double short time auto-correlation is not clearly explained, either. What is the input and what is the output?

Response 6: Thanks for the reviewer's questions and comments. In our previous research work (literature [25]), we have analyzed the effect of the number of input feature images on the signal recognition rate. We found that increasing the number of input feature images and combining multiple parallel CNNs can effectively improve the classification accuracy of the model. Based on this research conclusion, in the research of this manuscript, we still hope to get multiple feature images that can characterize the signal to improve the classification accuracy of the network. The above is the motivation for this manuscript to adopt double short-time autocorrelation. As a modification, this article has explained the motivation at the position of Section 3.3. For the specific implementation of double short-time autocorrelation, the input is also a signal, and then the signal is short-time autocorrelated. The difference between double short-time autocorrelation and short-time autocorrelation is that each autocorrelation sequence is autocorrelated again, and then the same clip(·) function and splice function are used to realize the splicing of autocorrelation sequence. The splicing result is the output of double short-time autocorrelation. ( lines 306 - 317 )

Point 7: Two structures are given in Figure 13. How can we fit these two structures in the framework in Fig. 1?  What is the purpose of the path "Training data pre-training" in Fig. 1?  Two window functions are mentioned and compared. Which one is then finally applied in the algorithm?

Response 7: Thanks for the reviewer's questions. In Figure 1, this manuscript has framed a classifier module with a dashed line. It consists of three sets of parallel neural networks. This classifier module is the part that should be replaced by the two structures in Figure 12. For a deep learning classifier, a certain amount of data should be used for training first, and then the trained classifier should use test data to evaluate the performance of the classifier. The "Training data pre-training" in Figure 1 means that the generated feature images are used as training data to train the classifier. The original block diagram does have some problems and has been modified. ( lines 118 - 119 )

Point 8: Two window functions are mentioned and compared. Which one is then finally applied in the algorithm?

Response 8: Thanks for the reviewer's questions. For the window function mentioned in the article, the Hamming window is the final window function used in the study, and the Kaiser window is mainly used for comparison. The selection of the window function has been further explained in the Section 3.3. ( lines 333 - 339 )

Point 9: Line 85, a right bracket is redundant. 

Response 9: Thanks for the reviewer's reminder, this is indeed a mistake in the manuscript. We have modified according to reviewer's comments. ( lines 71 - 72)

Point 10: What does TFI stand for ?

Response 10: Thanks for the reviewer's reminder, this is indeed a mistake in the manuscript. We have modified according to reviewer's comments. ( line 18 )

TFI:  The recent LPI radar waveform recognition technique in utilizes CWD to generate two-dimensional time-frequency image (TFI) for an input to the CNN based classification technique.

Point 11: The notations used in Table 1 are not defined before using Line 167, equation (2). I assume that the phase should be generally the integral of a time varied frequency function over the time interval.

Response 11: Thanks for the reviewer's reminder, φ(k) represent the instantaneous phase offset of the signal. To avoid ambiguity, we have adjusted the order of the paragraphs. ( lines 121 - 141 )

Point 12: Regarding plots in Fig. 3, why do the plots with worse SNR (-9 dB) in the second row seem to have a better contrast than those in the first row with SNR = 0 dB?

Response 12: Thanks for the reviewer's reminder, this is indeed a mistake in the manuscript. We have modified according to reviewer's comments and integrated Figure 3 and Figure 4 to make the comparison of the two window functions. ( lines 187 - 189 )

Point 13: Line 340, Equaiton (18) does not exist.

Response 13: Thanks for the reviewer's reminder, this is indeed a mistake in the manuscript. We have modified according to reviewer's comments. (line 318 )

Reviewer 3 Report

there are many typos in the text, such the use of capitalize after a comma or an space (L34 "Therefore, In order....", L44 "radar emitter signals; For another...", L166 "can be Computational expressed..."...) the authors repeat the same word, they should use synonimous: L10 "intercepted noisy ... of intercept", L19 "hybrid model classifier for classification", other typos: L85 "LFM), NCPM..." grammatical errors: "L112 we proposes" the introduction and other parts of the manuscript are hard to follow, with excessive text which does not provide important information to the reader. A clear case is the paragraph L60 to 107. The text must be rewritten, and focused on the line of the manuscript also in the introduction, the explanation of the objectives of the analysis should be simplified (L108 to 127), because they are not really clear after reading this long paragraph. The reader needs to see easily which are the goals after five or six lines, at least. in some case the authors do not provide a clear explanation of why they choose a technique or another. This is confusing, at least, for a people not expertise in the subject all the figures need a color scale Figure 4 text should be changed: "the same that fig. 3, but for the Hamming window" L229: they use s(t) for defining noise and signal Conclusions are so vague, after 21 pages of manuscript

Author Response

Response to Reviewer 3 Comments

Point 1: There are many typos in the text, such the use of capitalize after a comma or an space (L34 "Therefore, In order....", L44 "radar emitter signals; For another...", L166 "can be Computational expressed..."...)

Response 1: Thanks for the reviewer's reminder, this is indeed a mistake in the manuscript. We have modified according to reviewer's comments. ( line 33, line 44, lines 125)

Point 2: The authors repeat the same word, they should use synonimous: L10 "intercepted noisy ... of intercept", L19 "hybrid model classifier for classification", other typos: L85 "LFM), NCPM..." grammatical errors: "L112 we proposes"

Response 2: Thanks for the reviewer's reminder, this is indeed a mistake in the manuscript. We have modified according to reviewer's comments. ( line 10, line 19, line 71, line 81 )

Point 3: The introduction and other parts of the manuscript are hard to follow, with excessive text which does not provide important information to the reader. A clear case is the paragraph L60 to 107. The text must be rewritten, and focused on the line of the manuscript also in the introduction, the explanation of the objectives of the analysis should be simplified (L108 to 127), because they are not really clear after reading this long paragraph. The reader needs to see easily which are the goals after five or six lines, at least.

Response 3: Thanks for the reviewer's reminder, we have modified and rewritten according to reviewer's comments. ( lines 62 - 80, lines 81- 92 )

Point 4: In some case the authors do not provide a clear explanation of why they choose a technique or another. This is confusing, at least, for a people not expertise in the subject.

Response 4: Thanks for the reviewer's reminder, we have modified according to reviewer's comments. We add the explanation of CWD for TFI ( lines 148 - 174 ), double short-time autocorrelation ( lines 306 - 317 ),  choice of window function ( lines 333 - 339 ).

Point 5: All the figures need a color scale.

Response 5: Thanks for the reviewer's reminder, we have modified according to reviewer's comments.( figure 2, figure 3, figure 4, figure 5, figure 7, figure 9, figure 10, figure 11 )

Point 6: Figure 4 text should be changed: "the same that fig. 3, but for the Hamming window".

Response 6: Thanks for the reviewer's reminder, we have modified according to reviewer's comments and integrated Figure 3 and Figure 4 to make the comparison of the two window functions. ( lines 187 - 189 )

Point 7: L229: they use s(t) for defining noise and signal.

Response 7: Thanks for the reviewer's reminder, this is indeed a mistake in the manuscript. We have modified according to reviewer's comments and used x (t) instead of s (t). ( lines 212 - 218 )

Point 8: Conclusions are so vague, after 21 pages of manuscript.

Response 8: Thanks for the reviewer's reminder, we have modified according to reviewer's comments and rewritten the Conclusions. ( lines 582 - 595 )

Round 2

Reviewer 2 Report

I am satisfied with the improvement performed in this revision. All my questions have been answered. There is only a concern about the "training data pre-training" in the Fig.1. I still have some questions on it:

Generally speaking, a classifier is first trained / optimized in a training phase with labelled training data. After finishing training the model / classifier, it is ready to perform inference, i.e., classify the test data. To which phase does this "training data pre-training" belong ?  If it is measure belonging to the inference phase in order to optimize the classification performance online, how to label the generated feature images which are used training data ?  How does this "training data pre-training" path exactly influence the three neural networks ? What mathematical model was applied there ?

Author Response

Response to Reviewer 2 Comments

Point 1: Generally speaking, a classifier is first trained / optimized in a training phase with labelled training data. After finishing training the model / classifier, it is ready to perform inference, i.e., classify the test data. To which phase does this "training data pre-training" belong ?  If it is measure belonging to the inference phase in order to optimize the classification performance online, how to label the generated feature images which are used training data ?  How does this "training data pre-training" path exactly influence the three neural networks ? What mathematical model was applied there ?

Response 1: Thanks for the reviewer's reminder, this is indeed a mistake in the manuscript. The reviewer mentioned "generally speaking, a classifier is first trained / optimized in a training phase with labelled training data. After finishing training the model / classifier, it is ready to perform information, i.e., classify the test data", all of these are exactly what we want to express. In this manuscript we use "training data pre-training" to express this meaning. In the first reply, we have explained, but the explanation may still not be accurate enough. Therefore, as a modification, the manuscript has replaced "training data pre-training" in Fig. 1 with "training stage". We believe that this expression will allow readers to better understand with Figure 1. ( line 118)